# BézierSeg: Parametric Shape Representation for Fast Object Segmentation in Medical Images

**DOI:** 10.3390/life13030743

**Published:** 2023-03-09

**Authors:** Haichou Chen, Yishu Deng, Bin Li, Zeqin Li, Haohua Chen, Bingzhong Jing, Chaofeng Li

**Affiliations:** 1Collaborative Innovation Center for Cancer Medicine, State Key Laboratory of Oncology in South China, Guangdong Key Laboratory of Nasopharyngeal Carcinoma Diagnosis and Therapy, Sun Yat-sen University Cancer Center, Guangzhou 510060, China; 2Department of Information, Sun Yat-sen University Cancer Center, Guangzhou 510060, China; 3School of Electronics and Information Technology (School of Microelectronics), Sun Yat-sen University, Guangzhou 510275, China

**Keywords:** Bézier curves, deep learning, object segmentation

## Abstract

Background: Delineating the lesion area is an important task in image-based diagnosis. Pixel-wise classification is a popular approach to segmenting the region of interest. However, at fuzzy boundaries, such methods usually result in glitches, discontinuity or disconnection, inconsistent with the fact that lesions are solid and smooth. Methods: To overcome these problems and to provide an efficient, accurate, robust and concise solution that simplifies the whole segmentation pipeline in AI-assisted applications, we propose the BézierSeg model which outputs Bézier curves encompassing the region of interest. Results: Directly modeling the contour with analytic equations ensures that the segmentation is connected and continuous, and that the boundary is smooth. In addition, it offers sub-pixel accuracy. Without loss of precision, the Bézier contour can be resampled and overlaid with images of any resolution. Moreover, clinicians can conveniently adjust the curve’s control points to refine the result. Conclusions: Our experiments show that the proposed method runs in real time and achieves accuracy competitive with pixel-wise segmentation models.

## 1. Introduction

Image segmentation is a fundamental task in medical image processing. According to the statistics of the Grand Challenges competition in biomedical image analysis, there were 10, 14 and 13 tasks related to image segmentation in 2018, 2019 and 2020, respectively [1]. Deep learning has achieved huge success in image processing due to its accuracy and generality [2,3,4,5,6]. That is why most biomedical annotation tasks employ deep learning segmentation models such as U-Net [7], SegNet [8], DeepLab v3+ [9], etc. However, these pixel-wise deep learning segmentation models may not fully satisfy biomedical annotation problems due its own specialty: Firstly, most targets of the biomedical segmentation problem are solid objects, such as skin lesions, which have continuous and confined borders. Secondly, biomedical annotation tasks are normally not final tasks, but an intermediate step of the whole therapy process. Further manipulations might be applied to the annotations for down-steam tasks such as diagnosis, lesion measurement, radiotherapy, etc. As for pixel-wise algorithms, the output may not have a clear contour, or it may have burrs along the contour. Of course, we can do extra processing, such as binarizing the segmentation heat map and passing it to OpenCV [10] to find the contour. However, the output might contain multiple contours or a too complex contour for a single object. The first case needs additional post-processing such as NMS to obtain the most possible contour, and the second case needs a contour simplification algorithm to obtain a reasonable contour for clinicians to further process. These drawbacks show that the pixel-wise segmentation models might not be suitable for biomedical annotations. Moreover, all pixel-wise segmentation models need upsampling operations that further slow down the whole process. In application, the model runs in a backend server, and the results are transmitted to a front-end client, such as the web. Transmitting pixel-wise map data does not use bandwidth economically. To overcome these problems and to provide an efficient, accurate, robust and concise solution that simplifies the whole segmentation pipeline in AI-assisted applications, we propose a contour-based model—BézierSeg, an end-to-end segmentation model that does not need upsampling operations and can output a clear Bézier contour directly. The Bézier curve has been widely used in enterprise designs because of its user-friendly properties. Therefore, the predicted Bézier contour of the lesion area can be easily manipulated by clinicians for further study. Our main contributions are threefold:We propose to use parametric curves for shape encoding and reframe the pixel-wise classification problem into a point coordinate autoregression problem, thus providing convenience for many practical applications in clinical scenarios, e.g., manual refinement of prediction and data transmission.We propose BézierSeg, an end-to-end solution that can directly output the control points of the Bézier curves that encompass the detected object. We also devise a Bézier Differentiable Shape Decoder (BDSD) that further improves the segmentation performance.We validate our model on three medical image datasets. Experimental results show that BézierSeg can reach comparable results to the mainstream pixel-based solution while achieving 98 frames per second on a single Tesla V100 GPU for object segmentation.

The various parts of this paper are organized as follows: Section 2 provides an introduction and background knowledge of related work, and discusses the current common pixel-wise and contour-based segmentation algorithms. The definition and implementation of the Bézier curve-based segmentation model will be given in Section 3. Section 4 includes implementation details, experimental results and evaluation. Finally, the limitations of BézierSeg and future work are discussed, and conclusions are drawn.

## 2. Related Works

In deep learning, segmentation usually includes semantic segmentation and instance segmentation. The aim of semantic segmentation is to give pixel-wise classification results for the whole input image, whereas instance segmentation outputs bounding boxes for the detected objects and pixel-wise segmentation occurs within the bounding boxes.

### 2.1. Pixel-Wise Segmentation

Most pixel-wise semantic segmentation models are fully convolutional networks (FCN), such as U-Net, PSPNET [11], BiSeNet [12] and DeepLab v3+. U-Net has been widely used in biomedical segmentation problems. It consists of a contracting part and an expanding part, forming a u-shaped architecture. The contracting path extracts context information, and the expansive path recovers the resolution. To reduce the loss of contextual information between sub-regions for scene parsing tasks, PSPNet incorporates suitable global contextual features by the proposed pyramid pooling module, which captures contextual relationships and gathers global information in a hierarchical way. The author of BiSeNet proposed to implement the functions of spatial information preservation and the receptive field offered by the Spatial Path and Context Path, respectively. The features of these two paths are fused by a Feature Fusion Module. The whole model can reach real-time inference speeds. DeepLab v3+ is a cutting-edge semantic segmentation model developed by Google, which employs the atrous spatial pyramid pooling (ASPP) layer to exploit multi-scale features. Two-stage methods perform instance segmentation by detecting bounding boxes, followed by pixel-level segmentation within each bounding box; for example, the well-known Mask R-CNN is a two-stage method [13]. It first detects objects and then uses a mask branch and RoI-Align to segment instances within the proposed boxes. To better exploit the spatial information inside the box, PANet [14] introduces bottom-up path augmentation, adaptive feature pooling and fuses mask predictions from fully connected layers and convolutional layers. Such two-stage approaches achieve a state-of-the-art performance. One-stage instance segmentation methods are free of region proposals. In these methods, models output the pixel-wise auxiliary information, and then a clustering algorithm groups information into object instances. Deep Watershed Transform [15] predicts the energy map of the whole image and uses the watershed transform algorithm for grouping object instances. YOLACT [16] generates prototype masks and the linear combination coefficients for each instance, and then linearly combines the prototype masks by corresponding coefficients to predict the instance-level mask.

### 2.2. Contour-Based Segmentation

PolarMask [17] uses polar representation to model the object boundary. The model is trained to simultaneously regress the object centroid as well as the length of 36 rays emitting uniformly with the same angle interval from the object centroid. Combined with the proposed Polar IoU Loss, several regression targets can be trained as a whole, and thus improve the segmentation result. Instead of directly regressing the distance of the rays, ESE-Seg [18] proposes to further convert the polar coordinates of the object boundary to an explicit shape encoding using Chebyshev polynomial fitting, and turns the regression target into the coefficients of the Chebyshev polynomial. Although polar coordinates are inherently more suitable for modeling circular object boundaries, both PolarMask and ESE-Seg model the radial distance as a single-valued function of the polar angle, which makes it impossible to model the case of the ray having multiple points that intersect with the object boundary for a given angle. Curve-GCN [19] is designed for assisting with the manual annotation of class agnostic objects; it parametrizes object boundaries with either polygons or splines, allowing annotation for both line-based and curved objects. They use a graph convolutional network (GCN) to predict all the vertices of the polygon or control points of the spline along the object boundary in an iterative inference scheme. DeepSnake [20] exploits the special topology of the object boundary as prior knowledge, and adopts circular convolution to predict the vertices along the object boundary. In order to obtain a precise object boundary, both GCN and Deep Snake have to run inference multiple times.

## 3. Proposed Approach

### 3.1. Parametric Representation

Parametric equations are commonly used to express the coordinates of points that make up a geometric object, such as a curve or surface. The general form of a parametric equation in two dimensions is shown in Equation (1):(1)$$\left\{\begin{array}{l} x=f\left(t\right)\\ y=g\left(t\right) \end{array}\right.$$
where $t\in \left[0,1\right]$ is the parameter and $f\left(t\right),g\left(t\right)$ are any explicit function of t. To recover the object shape, one can sample t from the domain of the equation and obtain both x and y according to Equation (1). Different from using a single-valued function to model the object shape, a parametric equation expresses the x and y coordinates of the object shape independently, which allows it to mimic a multi-valued function. This property provides much flexibility compared to the single-valued function for shape representation.

The Bézier curve is a set of parametric curves that have been widely used in many fields as an efficient design tool. The explicit definition of a Bézier curve can be expressed as Equation (2):(2)$$B\left(t\right)=\sum_{i=0}^{n}\left(\begin{array}{c} n\\ i \end{array}\right){\left(1-t\right)}^{n-i}t^{i}P_{i},0\leq t\leq 1$$
where $\left(\begin{array}{c} n\\ i \end{array}\right)$ is the binomial coefficients, $P_{i}=\left(x_{i},y_{i}\right)$ is the i-th control point of the Bézier curve and n is the degree of the Bézier curve. One can construct a Bézier curve by following these steps:

(1)Create the control polygon of the Bézier curve by connecting the consecutive control points.(2)Insert intermediate points to each line segment, with the ratio $t:\left(1-t\right)$.(3)Treat the intermediate points as the new control points, and repeat step (1) and (2) until you end up with a single point.(4)As t varies from 0 to 1, the trajectory of that single point forms the curve.

Figure 1 shows this process for constructing a cubic Bézier curve. The use of a Bézier curve can reduce the number of parameters for shape encoding. As shown in Figure 2, although the curve is determined by only four control points, it can guarantee the shape quality since the precision of the curve representation fully depends on how densely you sample ts within the range of $\left[0,1\right]$, whereas the polygon representation requires more vertices to achieve similar precision.

Note that one can model the object shape using a Bézier curve with polar coordinates as well. In order to make the manual post-refinement of the segmentation result much easier, we instead model the object shape with Cartesian coordinates. We conducted an experiment to study the accuracy of curve regression with respect to the degree of the Bézier curve, and, according to Table 1, we choose n=5 for a trade-off between usability and segmentation accuracy. To retain a higher accuracy after converting the object boundary to the Bézier curve representation, we propose to parametrize the object shape with piece-wise Bézier curves. Specifically, we first split the whole object boundary by its four extreme points, i.e., the top, the leftmost, the bottom and the rightmost points of the object. Then, for each part of the object boundary, we parametrize it with a Bézier curve. Figure 3a gives an example of our Bézier curve representation. We also compare our Bézier curve representation with the Chebyshev polynomial shape encoding proposed in [18]. Using Figure 3b, we found that our Bézier curve representation can avoid the oscillation at the end of the object boundary in contrast to the Chebyshev polynomial shape encoding.

### 3.2. Model Architecture

As shown in Figure 4, we adopt ResNet-101 [21] as the backbone of our model. Since our model is free of up-sampling layers, we simply remove the softmax activation layer commonly used in classification tasks [22,23,24] and add an extra fully connected layer to map the features extracted by the backbone network to the coordinate predictions. The number of output nodes in the last fully connected layer matches the regression targets. For example, if $n_{e}$ extreme points and $n_{c}$ control points are needed, the output of the last fully connected layer should be $\left(n_{e}+n_{c}\right)\times 2$ nodes, without any activation layers. During training, the Bézier Differentiable Shape Decoder takes the predicted control point coordinates and ground truth control point coordinates as input, and outputs two sets of sampled point coordinates controlled by its two inputs. Reconstruction loss is built on these two sets of sampled point coordinates; minimizing this loss term indirectly helps in learning the control point coordinates. Note that the idea in this paper can be easily applied to the mainstream object detection frameworks, which provide an alternative for instance segmentation.

### 3.3. Ground Truth Label Generation

Given a mask of an object, we first extract the object boundary from the mask. Then, we find four extreme points of the object and use them to split the whole shape into four parts. If there exist multiple extreme points, such as two or more top points, we use the top left corner point, which is the same as the bottom left one, the bottom right one and the top right one for the leftmost, bottom and rightmost extreme point, respectively. For each part of the object boundary, we fit it with a fifth-order Bézier curve. The first and last control points of the Bézier curve are set to be the beginning point and end point of that piece of the shape, which are two consecutive extreme points as well. The four intermediate control points of the Bézier curve remain unknown. We convert Equation (2) to the matrix form and place the coordinates of the object boundary part into the equation, as shown in Equation (3):
(3)$$\left[\begin{array}{ccccc} C_{1}^{1}, & C_{1}^{2}, & C_{1}^{3}, & \cdots , & C_{1}^{n}\\ C_{2}^{1}, & C_{2}^{2}, & C_{2}^{3}, & \cdots , & C_{1}^{n}\\ C_{3}^{1}, & C_{3}^{2}, & C_{3}^{3}, & \cdots , & C_{3}^{n}\\ \vdots & \vdots & \vdots & \ddots & \vdots \\ C_{m}^{1}, & C_{m}^{2}, & C_{m}^{3}, & \cdots , & C_{m}^{n} \end{array}\right]\left[\begin{array}{cc} x_{1}, & y_{1}\\ x_{2}, & y_{2}\\ x_{3}, & y_{3}\\ \vdots & \vdots \\ x_{n}, & y_{n} \end{array}\right]=\left[\begin{array}{cc} {\hat{x}}_{1}, & {\hat{y}}_{1}\\ {\hat{x}}_{2}, & {\hat{y}}_{2}\\ {\hat{x}}_{3}, & {\hat{y}}_{3}\\ \vdots & \vdots \\ {\hat{x}}_{m}, & {\hat{y}}_{m} \end{array}\right]$$
in which m represents the total number of points of the object boundary part, n represents the number of control points of the Bézier curve, $\left(x_{i},\;y_{i}\right)$ is the i-th control point, $\left({\hat{x}}_{i},{\hat{y}}_{i}\right)$ is the i-th points of the object boundary part and $C_{i}^{j}=\left(\begin{array}{c} n\\ j \end{array}\right)(1-{\left.t_{i}\right)}^{n-j}t_{i}^{j},t_{i}\in \left[0,1\right]$. Inspired by De Casteljau’s algorithm for constructing a Bézier curve, we assume that $t_{i}=\left(i-1\right)/\left(m-1\right)$ for the i-th point of the object boundary part. Since the first and the last row of Equation (3) are identity expressions representing the extreme points, and the first and last column are constant values, we remove the first and the last row from Equation (3) and subtract the constant values from both side of Equation (3). To simplify the notation, let the following:$$A=\left[\begin{array}{ccccc} C_{2}^{2}, & C_{2}^{3}, & C_{2}^{4}, & \cdots , & C_{2}^{n-1}\\ C_{3}^{2}, & C_{3}^{3}, & C_{3}^{4}, & \cdots , & C_{3}^{n-1}\\ C_{4}^{2}, & C_{4}^{3}, & C_{4}^{4}, & \cdots , & C_{4}^{n-1}\\ \vdots & \vdots & \vdots & \ddots & \vdots \\ C_{m-1}^{2}, & C_{m-1}^{3}, & C_{m-1}^{4}, & \cdots , & C_{m-1}^{n-1} \end{array}\right],\;b=\left[\begin{array}{cc} \hat{x_{2}}-x_{1}C_{2}^{1}-x_{n}C_{2}^{n}, & \hat{y_{2}}-y_{1}C_{2}^{1}-y_{n}C_{2}^{n}\\ \hat{x_{3}}-x_{1}C_{3}^{1}-x_{n}C_{3}^{n}, & \hat{y_{3}}-y_{1}C_{3}^{1}-y_{n}C_{3}^{n}\\ \hat{x_{4}}-x_{1}C_{4}^{1}-x_{n}C_{4}^{n}, & \hat{y_{4}}-y_{1}C_{4}^{1}-y_{n}C_{4}^{n}\\ & \vdots \\ \vdots & \vdots \\ x_{m-1}-x_{1}C_{m-1}^{1}-x_{n}C_{m-1}^{n}, & y_{m-1}-y_{1}C_{m-1}^{1}-y_{n}C_{m-1}^{n} \end{array}\right],\;c=\left[\begin{array}{cc} x_{2}, & y_{2}\\ x_{3}, & y_{3}\\ x_{4}, & y_{4}\\ \vdots & \vdots \\ x_{n-1}, & y_{n-1} \end{array}\right]$$

Thus, Ac=b. By computing the pseudo-inverse of A and multiplying it by the right side of the equation, we can obtain the coordinates of the intermediate control points $c=A^{-1}b$. Finally, we concatenate the coordinates of 4 extreme points and 16 intermediate control points as our regression target ($n_{e}=4,\;n_{c}=16$), resulting in a 40-dimensional vector for each input image. Similar to ESE-Seg, we also conduct a sensitivity analysis for our Bézier curve representation. Specifically, we randomly sample some noises from $N\left(0,\delta \right)$ and add them to the control points and the extreme points to imitate the uncertainty behavior of the convolutional neural network. We compare our Bézier curve representation with the polygon shape representation. To maintain the same complexity for both shape representations, we evenly select 20 points along the object boundary for the polygon representation. Figure 5 shows the robustness of our Bézier curve representation. As shown in Figure 5, the Bézier curve representation always achieves a higher mean intersection-over-union (MIOU) than the polygon representation with the same number of points. We can conclude that the Bézier curve representation is superior to the polygon representation because it allows us to describe a curve more precisely. The Bézier curve representation is robust when used on the endoscopy images of upper gastrointestinal cancers (EIUGC) dataset and the international skin imaging collaboration skin lesion challenge (ISIC) dataset, remaining above a 0.6 MIOU even when the noise of δ=20 was injected. For the same perturbation, the MIOU of the nasopharyngeal carcinoma magnetic resonance imaging (NPCMRI) dataset dropped rapidly, mainly due to its smaller average size of ROI compared to the other two datasets.

### 3.4. Bézier Differentiable Shape Decoder

Similar to FourierNet [25], we devise a novel differentiable shape decoder named the Bézier Differentiable Shape Decoder, abbreviated as BDSD. FourierNet uses Inverse Fast Fourier Transformation as the shape decoder to convert the coefficients of the Fourier series into contour points, whereas we apply the parametric equation of Bézier curves to map the control points to contour points. Following Equation (2), we randomly sample N ts from $U\left[0,1\right]$ and use the same ts to reconstruct N ground truth boundary points and the corresponding predicted boundary points. In all experiments, we set N=72. The BDSD module is fully differentiable and allows the gradient to back propagate through the network, thus providing extra supervision during the training phase.

### 3.5. Loss

The overall loss function for training our model contains two individual loss terms. The first one, $L_{ce},$ is the smooth L1 loss for the regression of control points and extreme points, and the second one, $L_{matching}$, is another smooth L1 loss for point-matching learning, which measures the differences between the outputs of the BDSD module. $L_{ce}$ weights each control point equally and $L_{matching}$ implicitly assigns different weights to the control points. Equation (4) shows the overall loss function:(4)$$L=\lambda_{ce}L_{ce}+\lambda_{matching}L_{matching}\;\;\;\;$$
where $\lambda_{ce}$ and $\lambda_{matching}$ are balancing hyper parameters; we set both of them equal to 1 in all experiments.

## 4. Experiments

### 4.1. Datasets and Evaluation Metric

The endoscopy images of upper gastrointestinal cancers (EIUGC) dataset is an upper gastrointestinal endoscopy image dataset that aims to detect upper gastrointestinal cancers. The EIUGC dataset is a collection of 38,453 endoscopy images. The border of each cancer lesion is marked by highly experienced endoscopists from SYSUCC. We separated the dataset into a training set, a validation set and a test set with 30762, 3845 and 3846 images, respectively. The nasopharyngeal carcinoma magnetic resonance imaging (NPCMRI) dataset enrolled 375 patients at SYSUCC from January 2012 to December 2014. The NCMRI dataset contains multi-parametric MRI sequences of nasopharyngeal carcinoma patients, including T1w, T2w and T1c. Clinicians delineated the invasive ROIs of NPC in T1c by referencing T1w and T2w sequences. As a result, we used the T1c sequence as source images. In total, 2,337 images were collected. 

The international skin imaging collaboration (ISIC) skin lesion challenge dataset of 2018 was released during the ISIC Challenge, with the aim of skin lesion analysis towards melanoma detection. One of its three tasks is lesion segmentation. The dataset includes 2,576 skin images and their corresponding masks, of which 2060 are in the training set, 258 in the validation set and 258 in the test set. All ground truth data were reviewed and curated by practicing dermatologists with expertise in dermoscopy. An image may be associated with multiple mask annotations. In that case, we randomly selected one of the expert annotations and used novice annotation if no expert annotation was provided. The above three datasets are summarized in Table 2.

We use the MIOU, Hausdorff distance [26], Matthews correlation coefficient [27] (mcc), area under the curve (auc), false-positive rate (fp) and false-negative rate (fn) to evaluate the proposed method, a contour-based model, PolarMask ResNet101 and a pixel-wise model, DeepLab v3+ ResNet101, on the EIUGC, NPCMRI and ISIC datasets.

### 4.2. Implementation Details

For training DeepLab v3+ ResNet101, we use the binary cross-entropy loss function and initialize the model with the weights pre-trained on COCO [28] train2017, and for training BézierSeg, we initialize the backbone of our model with the weights pre-trained on ImageNet [29]. As for the configuration of PolarMask, we use ResNet101 as the backbone and use 36 rays for shape representation. During training, for the EIUGC and the ISIC datasets, we use data augmentation techniques such as random horizontal flipping or vertical flipping [30,31] with a probability of 0.5, image rotation with a random angle of $\left[-20,20\right]$, and image scaling with a random scale factor of $\left[0.8,1.25\right]$ followed by random fixed size image cropping. For the NPCMRI dataset, we apply the same techniques above except that we replace the random image scaling and cropping with the fixed image resizing augmentation, and change the range of the rotation angle from $\left[-20,20\right]$ to $\left[-10,10\right]$. During the inference phase, we simply resize the input images to the size of $\left(256,256\right)$. For both training and inference phases, we normalize the input images by dividing their pixel value by 255. We dynamically generate both the extreme points and the control points after performing data augmentation. 

For the ISIC dataset, we apply morphological transformations to the mask to remove the spikes along the object boundary before fitting the Bézier curves. The initial learning rate is set to $1\times 10^{-3}$ for all experiments. We also adopt an adaptive learning rate scheduling strategy during training. Specifically, we reduce the learning rate by half when the loss of the validation set has stopped decreasing for 15 epochs and the monitoring threshold is set to $1\times 10^{-2}$ and 0.5 for DeepLab v3+ ResNet101 and BézierSeg, respectively. We train the models for 100 epochs in all experiments and evaluate the performance on the validation set at the end of each epoch. We pick up the model with the highest MIOU on the validation set to evaluate the performance on the hold-out test set. The final results are obtained by running the experiments three times with different random seeds and taking the average.

## 5. Results and Discussion

### 5.1. Quantitative Evaluation

As shown in Table 3, the segmentation performance of BézierSeg 50 is slightly worse than BézierSeg 101; therefore, we use ResNet-101 as the backbone of our model by default. Without any bells and whistles, BézierSeg can achieve a competitive result with pixel-wise methods. On both the EIUGC and ISIC datasets, the differences in the MIOU between these two methods are less than 0.02. The average ROI sizes in the EIUGC, NPCMRI and ISIC datasets are 29,337 pixels, 14,152 pixels and 745 pixels, respectively. The proposed BDSD module effectively improves the performance, especially on datasets with a smaller ROI size. The reason may be that objects with a small size have a greater curvature, making the loss function give more weights to the relevant control points during training.

BézierSeg outperforms DeepLab v3+ ResNet101 on the NPCMRI dataset by 21% and behind DeepLab v3+ on EIGUC and ISIC by only 1.8% and 0.6%, respectively. In terms of distance, BézierSeg outperforms DeepLab v3+ on both the ISIC and NPCMRI datasets. PolarMask performs relatively poorly on all three datasets.

From Figure 6 we can see that when the ROI size is greater than 20k, DeepLab v3+ ResNet101 performs slightly better; when the ROI size is between 4k and 20k, the proposed model and DeepLab v3+ ResNet101 perform similarly; and when the ROI size is less than 4k, both BézierSeg and PolarMask outperform DeepLab v3+ ResNet101. The overall performance of BézierSeg is less sensitive to the size of ROI comparing to DeepLab v3+ ResNet101. 

We also compare the FPS between BézierSeg, PolarMask and DeepLab v3+ on the same machine equipped with one Tesla V100 graphics card. As shown in Table 4, BézierSeg, which does not require upsampling layers, has 42.6 M parameters while DeepLab v3+ ResNet101 has 58.6M. With a simpler pipeline, BézierSeg doubles the FPS compared to DeepLab v3+ ResNet101. Both BézierSeg and PolarMask are lightweight models with a similar architecture, and thus, they show no significant differences when running without post-processing. However, BézierSeg needs more time to reconstruct the smooth contour during the post-processing stage. The speed of BézierSeg makes it more suitable for real-time medical segmentation. A smaller size of model has more flexibility to transfer to an edge-computing device.

### 5.2. Qualitative Evaluation

As shown in Figure 7, BézierSeg can always output a smooth contour, whereas DeepLab v3+ ResNet101 outputs a rugged contour and often produces multiple separated contours. PolarMask gives polygonal outputs that roughly ensure smoothness. However, it is difficult to handle the dumbbell shape, and thus the model outputs a pebble shape in most of those cases. The use of parametric shape representation allows BézierSeg to predict object contours better than PolarMask, making BézierSeg less sensitive to local artifacts. In summary, BézierSeg can be used as an alternative to DeepLab v3+ ResNet101 while having a faster speed that is similar to PolarMask.

## 6. Conclusions

We propose the BézierSeg model, which can directly output control points of Bézier curves without post-processing. Its simple pipeline provides real-time computing speeds. In the experiments, it was found that BézierSeg shows competitive accuracy on multiple medical datasets. However, our model fails to handle disconnected areas such as dilated shape contours due to the limitation of the 2D Bézier curve representation. One possible solution in this case is to incorporate parametric surface representation, such as a level set. We will explore the potential application of the Bézier surface for 3D object segmentation in our future work, which is also a promising and practical segmentation solution for three-dimensional medical data such as computed tomography (CT) and magnetic resonance (MR) images.

## Figures and Tables

**Figure 1 life-13-00743-f001:**
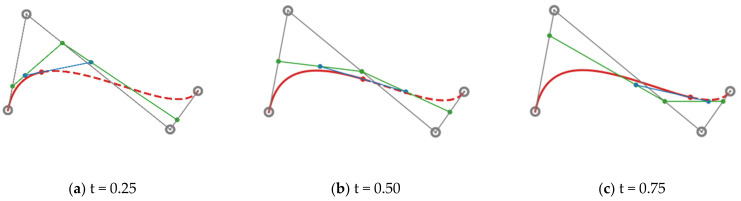
The process of constructing a cubic Bézier curve. The trajectory of the red dot forms the curve as t varies from 0 to 1.

**Figure 2 life-13-00743-f002:**
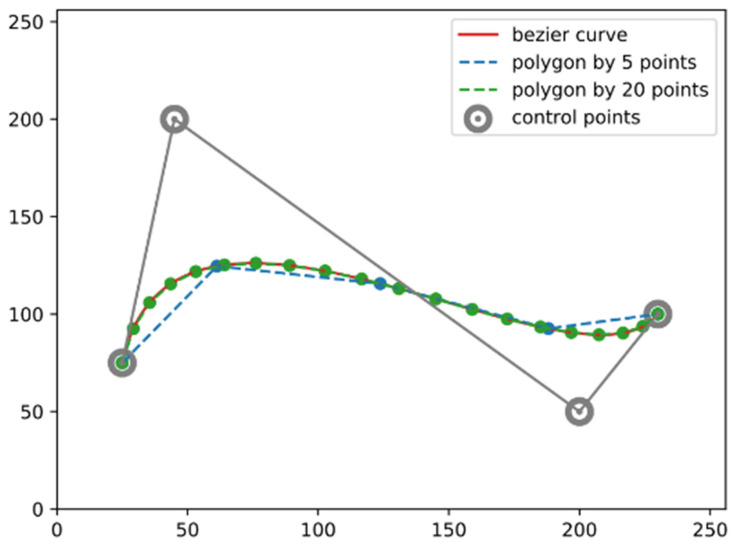
Bézier curve vs. polygon. The polygon representation requires more vertices in order to precisely describe the shape.

**Figure 3 life-13-00743-f003:**
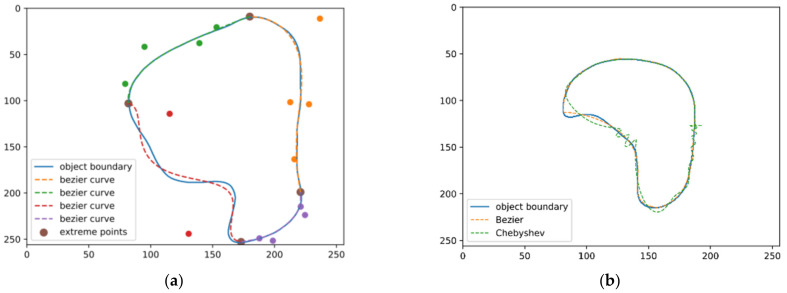
(**a**) Our piece-wise Bézier curve representation, the colored points are the control points of each corresponding curve segment. (**b**) Our Bézier curve representation can avoid the oscillation at the end of the object boundary in contrast to Chebyshev polynomial shape encoding.

**Figure 4 life-13-00743-f004:**
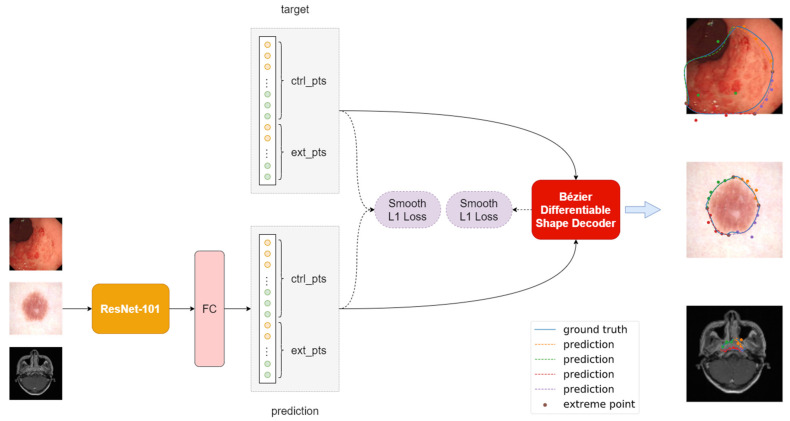
Model architecture ext_pts and ctrl_pts are the coordinates for $n_{e}$extreme points and $n_{c}$ control points. The colored points rendered on the images are the control points of each corresponding curve segment. Image features extracted from the ResNet-101 are passed to the fully connected FC layer to obtain $\left(n_{e}+n_{c}\right)\times 2$ outputs. Thus, the predicted coordinates of ext_pts and ctrl_pts regress to the target ext_pts and ctrl_pts supervised by smooth L1 loss, denoted as $L_{ce}$. The Bézier Differentiable Shape Decoder module reconstructs N predicted boundary points and N ground truth boundary points. $L_{matching}$ is the smooth L1 distance (see Section 3.4 and Section 3.5).

**Figure 5 life-13-00743-f005:**
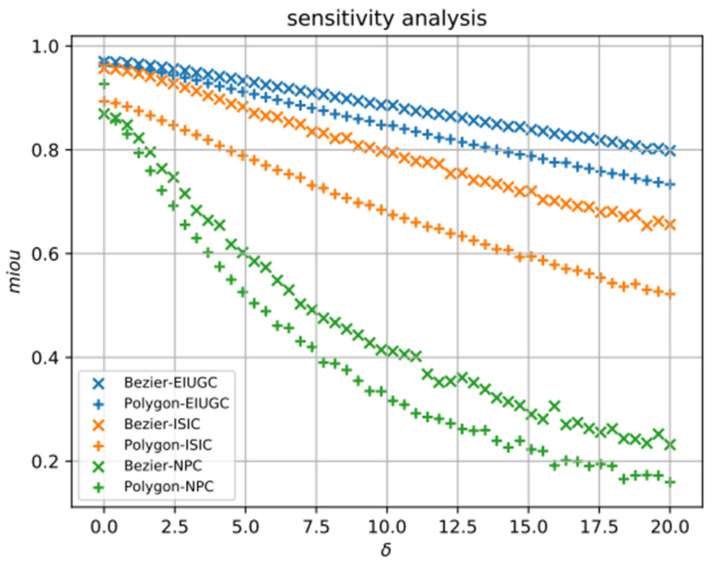
Sensitivity analysis. The result of the analysis indicated that the Bézier curve representation is more robust than polygon representation under different noise levels.

**Figure 6 life-13-00743-f006:**
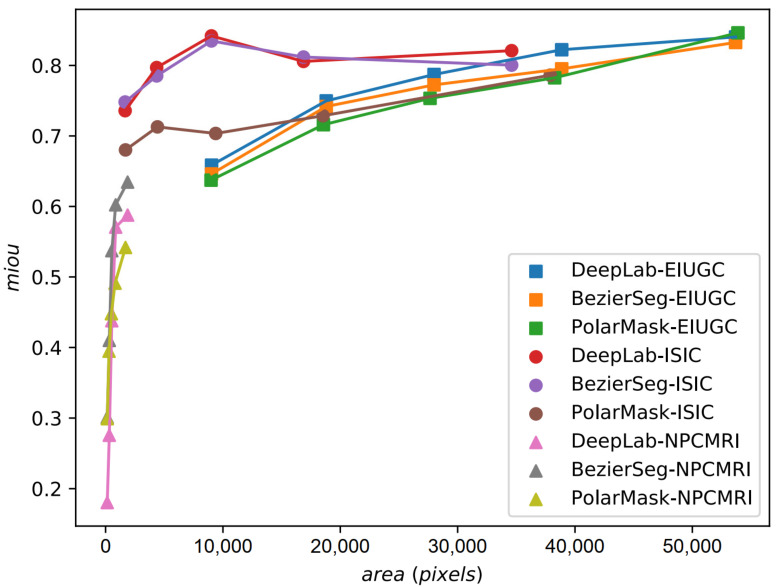
Performance comparison between BézierSeg, PolarMask and DeepLab v3+ on different area ranges.

**Figure 7 life-13-00743-f007:**
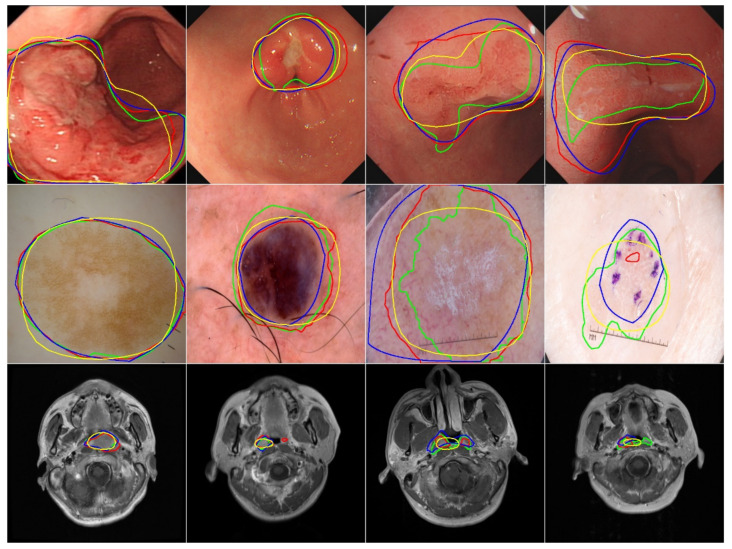
Qualitative result comparison between BézierSeg, DeepLab v3+ ResNet101 and PolarMask on three datasets. The first row, second row and the last row show the results on the EIUGC dataset, the ISIC dataset and the NPCMRI dataset, respectively. Green: ground truth label; red: DeepLab v3+ ResNet101; blue: BézierSeg; yellow: PolarMask.

**Table 1 life-13-00743-t001:** Degree of Bézier curve. A higher degree of Bézier curve brings higher accuracy, while too high of a degree degrades the performance.

**Degree of Bézier Curve (*n*)**	**3**	**5**	**7**	**9**
MIOU	0.753	0.755	0.749	0.736

**Table 2 life-13-00743-t002:** Dataset overview. The MIOU and SIOU in the table represent the mean intersection-over-union and the standard deviation of the intersection-over-union between the Bézier curve representation and the ground truth mask, respectively.

Dataset	Trainset	Valset	Testset	MIOU	SIOU
EIUGC	30,762	3845	3846	0.970	0.019
NPCMRI	1869	234	234	0.869	0.056
ISIC	2060	258	258	0.957	0.021

**Table 3 life-13-00743-t003:** Comparison of BézierSeg ResNet50 (BézierSeg 50), BézierSeg ResNet101 (BézierSeg 101), DeepLab v3+ ResNet101 (DeepLab v3+) and PolarMask ResNet101 (PolarMask) on three datasets. Curve MIOU is the mean intersection-over-union between the prediction and the ground truth Bézier curve shape representation, and Mask MIOU is the mean intersection-over-union between the prediction and the ground truth mask. The -, ◦, ✓ symbols indicate that the module is not needed, the module is not used, and the module is used respectively.

Dataset	Model	BDSD	Curve MIOU	Mask MIOU	Hausdorff	Mcc	Auc	fp	fn
EIUGC	DeepLab v3+	-	-	**0.772**	**14.330**	**0.781**	**0.891**	0.104	0.115
	PolarMask	-	-	0.747	16.263	0.753	0.878	0.129	0.116
	BézierSeg 50	◦	0.750	0.745	15.473	0.752	0.876	0.118	0.129
	BézierSeg 50	✓	0.750	0.745	15.573	0.751	0.876	0.126	0.122
	BézierSeg 101	◦	0.761	0.757	14.898	0.763	0.882	0.113	0.124
	BézierSeg 101	✓	0.762	0.758	14.818	0.763	0.882	0.119	0.117
ISIC	DeepLab v3+	-	-	**0.801**	8.139	**0.877**	**0.939**	0.028	0.093
	PolarMask	-	-	0.723	8.707	0.812	0.903	0.039	0.156
	BézierSeg 50	◦	0.796	0.791	**7.339**	0.869	0.934	0.029	0.104
	BézierSeg 50	✓	0.799	0.793	7.349	0.872	0.936	0.029	0.099
	BézierSeg 101	◦	0.797	0.792	7.414	0.866	0.934	0.031	0.102
	BézierSeg 101	✓	0.801	0.796	7.406	0.870	0.936	0.030	0.097
NPCMRI	DeepLab v3+	-	-	0.413	9.553	0.671	0.820	0.003	0.357
	PolarMask	-	-	0.437	5.348	0.650	0.799	0.003	0.398
	BézierSeg 50	◦	0.482	0.467	5.285	0.685	0.840	0.003	0.317
	BézierSeg 50	✓	0.504	0.488	4.909	0.707	0.855	0.003	0.287
	BézierSeg 101	◦	0.500	0.478	5.258	0.696	0.856	0.004	0.284
	BézierSeg 101	✓	0.520	**0.499**	**4.886**	**0.717**	**0.866**	0.003	0.265

**Table 4 life-13-00743-t004:** Speed comparison between DeepLab v3+ Resnet101 (DeepLab v3+), BézierSeg Resnet101 (BézierSeg) and PolarMask Resnet101 (PolarMask), including both with/without post-processing to reconstruct the object boundary. The ◦, ✓ symbols indicate that not performing this operation and performing this operation respectively.

Model	Post-Processing	FPS
DeepLab v3+	◦	48.4
	✓	45.6
BézierSeg	◦	103.8
	✓	97.8
PolarMask	◦	103.3
	✓	101.9

## Data Availability

The authenticity of this article has been validated by uploading the key raw data onto the Research Data Deposit (RDD) public platform (www.researchdata.org.cn), with the approval RDD numbers RDDB2019000564 and RDDA2019001214 for the NPCMRI dataset and the EIUGC dataset, respectively.

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
