# Peer review of "BézierSeg: Parametric Shape Representation for Fast Object Segmentation in Medical Images"

_life, 2023, doi:10.3390/life13030743_

Round 1

Reviewer 1 Report

The article is devoted to developing the BezierSeg model, which displays Bezier curves covering the affected area of interest. The study's relevance is justified by delineating the affected area as an essential task in image-based diagnostics. Pixel classification is a popular approach to segmenting a region of interest. However, with indistinct boundaries, such methods usually result in glitches, discontinuity, or shutdown, which is inconsistent with the fact that the lesions are solid and smooth. To overcome these unwanted artifacts, the authors propose a BezierSeg model that outputs Bezier curves spanning the region of interest. Direct modeling of the contour using analytical equations ensures that the segmentation is connected and continuous and that the boundary is smooth. In addition, it offers sub-pixel precision. Without loss of precision, a Bezier path can be resampled and overlaid with images of any resolution. In addition, the clinician can conveniently adjust the curve's control points to refine the result. The experiments carried out as part of the study show that the proposed method works in real-time and achieves an accuracy comparable to pixel-based segmentation models.

Despite the satisfactory quality of the article, some shortcomings need to be corrected.

  1. The aim of the paper should be defined.
  2. It is recommended to expand the Related works section with extensions of described models (U-Net, PSPNET, etc.)
  3. The Bezier curve is a well-known approach. The authors' contribution should be highlighted.
  4. The selection of the model’s architecture, presented in Figure 4 should be justified and described in more detail in the text.
  5. It is recommended to describe the data used for the experimental investigation in more detail. The link to the dataset should be provided.
  6. It is recommended to include the Discussion section to compare the obtained results with other research conducted with the same dataset.
  7. The practical novelty of the research should be highlighted.
  8. Formulas are parts of sentences. The punctuation should be corrected.

In summarizing my comments, I recommend that the manuscript is accepted after major revision. 

Author Response

Q1: The aim of the paper should be defined.

Response: Thanks for your valuable suggestion. We have defined the aim of the paper as follows to make it more easily understood (Page 1, Line 13-15).

To overcome these problems and to provide an efficient, accurate, robust and concise solution that simplifies the whole segmentation pipeline in AI assisted applications, we propose the BézierSeg model which outputs Bézier curves encompassing the region of interest.

Q2: It is recommended to expand the Related works section with extensions of described models (U-Net, PSPNET, etc.)

Response: Thanks for your reminding. We have made changes. The detailed version is as follows (Page 2, Line 83-93).

  U-Net has been widely used in biomedical segmentation problems. It consists of a contracting part and an expansive part, forming a u-shaped architecture. The contracting path extracts context information and the expansive path recovers the resolution. To reduce the loss of contextual information between sub-regions for scene parsing tasks, PSPNet incorporates suitable global contextual features by the proposed pyramid pooling module, which captures contextual relationship and gathers global information in a hierarchical way. The author of BiSeNet proposed to implement the function of spatial information preservation and receptive field offering by Spatial Path and Context Path respectively, the features of these two paths are fused by a Feature Fusion Module. The whole model can reach real-time inference speed.

Q3: The Bezier curve is a well-known approach. The authors' contribution should be highlighted.

Response: Thank you for your careful work. We have highlighted our contribution as follows (Page 2, Line 55-68).

Our main contributions are threefold:

  1. We propose to use parametric curves for shape encoding and reframe the pixel-wise classification problem into point coordinate autoregression problem, thus providing convenience for many practical applications in clinical scenario, e.g., manual refinement of prediction, data transmission.
  2. We propose BézierSeg, an end-to-end solution that can directly output the control points of the Bézier curves that encompass the detected object. We also devise a Bézier Differentiable Shape Decoder (BDSD) that further improves the segmentation performance.
  3. We validate our model on three medical image datasets. Experimental result shows that BézierSeg can reach comparable result to the mainstream pixel-based solution while achieving 98 frames per second on single Tesla V100 GPU for object segmentation.

Q4: The selection of the model’s architecture, presented in Figure 4 should be justified and described in more detail in the text.

Response: Thank you for your careful work. We have described in more detail in the text. The detailed version is as follows (Page 5, Line 188-201).

As shown in Fig. 4, we adopt ResNet-101 [21] as the backbone of our model. Since our model is free of up sampling layers, we simply remove the softmax activation layer commonly used in classification tasks [22-24] and add an extra fully connected layer to map the features extracted by the backbone network to the coordinate predictions. The number of output nodes in the last fully connected layer matches the regression targets. For example, if  extreme points and  control points are needed, the output of the last fully connected layer should be  nodes, without any activation layer. During training, the Bézier Differentiable Shape Decoder takes the predicted control point coordinates and ground truth control point coordinates as input, and outputs two set of sampled point coordinates controlled by its two inputs. Reconstruction loss is built on these two set of sampled point coordinates, minimizing this loss term indirectly helps learning the control points coordinates. Note that the idea in this paper can be easily applied to the mainstream object detection frameworks, which provides an alternative for instance segmentation.

Q5: It is recommended to describe the data used for the experimental investigation in more detail. The link to the dataset should be provided.

Response: Thanks for your reminding.The EIUGC and NPCMRI datasets involve data security and our follow-up research, and we strive to disclose the corresponding links as soon as possible. This study focuses on demonstrating the generality and flexibility of our approach.

Q6: It is recommended to include the Discussion section to compare the obtained results with other research conducted with the same dataset.

Response: Thanks for your reminding. We have modified the Discussion section as follows to make it more appropriate(Page 10, Line 341-344; Page 12, Line379-384).

  1. Results and Discussion

5.1 Quantitative Evaluation

The average ROI sizes in EIUGC, NPCMRI and ISIC are 29337 pixels, 14152 pixels and 745 pixels respectively. The proposed BDSD module effectively improves the performance, especially on dataset with smaller ROI size. The reason may be that objects of small size have greater curvature, making the loss function give more weights to the relevant control points during training.

5.2 Qualitative Evaluation

However, it is difficult to handle the dumbbell shape and the model outputs pebble shape in most of that case. The use of parametric shape representation allows BézierSeg to predict object contours better than PolarMask, making BézierSeg less sensitive to local artifacts. In summary, BézierSeg can be used as an alternative to DeepLab v3+ ResNet101 while having a faster speed similar to PolarMask.

  1. CONCLUSION

Q7: The practical novelty of the research should be highlighted.

Response: Thank you for your careful work. We have highlighted our contribution as follows (Page 2, Line 55-68).

Our main contributions are threefold:

  1. We propose to use parametric curves for shape encoding and reframe the pixel-wise classification problem into point coordinate autoregression problem, thus providing convenience for many practical applications in clinical scenario, e.g., manual refinement of prediction, data transmission.
  2. We propose BézierSeg, an end-to-end solution that can directly output the control points of the Bézier curves that encompass the detected object. We also devise a Bézier Differentiable Shape Decoder (BDSD) that further improves the segmentation performance.
  3. We validate our model on three medical image datasets. Experimental result shows that BézierSeg can reach comparable result to the mainstream pixel-based solution while achieving 98 frames per second on single Tesla V100 GPU for object segmentation.

Q8: Formulas are parts of sentences. The punctuation should be corrected.

Response: Thanks for your reminding. We have modified the punctuation to make it more appropriate .

Reviewer 2 Report

The article entitled «BezierSeg: Parametric Shape Representation for Fast Object Segmentation in Medical Images » is well-written and, from my point of view, would be of interest for the readers of Life. In spite of this and before its publication I would recommend authors to perform the following changes:

Line 28: please introduce comma before ‘etc.’

The introductory section requires of a paragraph about the structure of the article and another paragraph that contains infomration about the aim of the research.

Line 101: when you refer to real numbers, please use the simbol commonly accepted for such kind of numbers and not simply R letter.

Figure 1: please, center it.

Figure 2: please enlarge it.

Figure 5: please, enlarge it.

Subsections of the article: in MPDI format letters are not employed for subsections numbering as for example 1.1, 1.2, etc. Pleae chek the whole manuscript and modify it.

Figure 6: please, enlarge it.

Table 4: I am not sure if such table is required ori f the information can be summarized in a short paragraph.

Finally, I am not sure if the font size of the references is the required by MDPI.

Author Response

-Reviewer 2

Q1: Line 28: please introduce comma before ‘etc.’

Response: Thanks for your reminding. We have modified the above statement as follows to make it more appropriate (Page 1, Line 30-32).

That is why most biomedical annotation tasks employ deep learning segmentation models like U-Net [7], SegNet[8], DeepLab v3+ [9], etc.

Q2: The introductory section requires of a paragraph about the structure of the article and another paragraph that contains infomration about the aim of the research.

Response: Thanks for your valuable suggestion. We have added the aim of the paper in the introduction section as follows (Page 2, Line 70-74,48-53).

The various parts of this paper are organized as follows: Section 2 provides an introduction and background knowledge of related work; discusses the current common pixel-wise and contour-based segmentation algorithms. The definition and implementation of the Bézier curve-based segmentation model will be given in Section 3. Section 4 includes implementation details, experimental results, and evaluation. Finally, the limitations of BézierSeg and future work are discussed, and conclusions are drawn.

Data of a pixel-wise map for transmitting is not bandwidth-economy. To overcome these problems and to provide an efficient, accurate, robust and concise solution that simplifies the whole segmentation pipeline in AI assisted applications, we propose a contour-based model - BézierSeg, an end-to-end segmentation model that does not need up sampling operations and can output a clear Bézier contour directly.

Q3: Line 101: when you refer to real numbers, please use the simbol commonly accepted for such kind of numbers and not simply R letter.

Response: Thanks for your reminding. We have modified the above statement as follows to make it more appropriate (Page 3, Line 136).

where  is the parameter and  are any explicit function of  .

Q4: Figure 1: please, center it.

Response: Thanks for your reminding. We have made changes.

Q5: Figure 2: please enlarge it.

Response: Thanks for your reminding. We have made changes.

Q6: Figure 5: please, enlarge it.

Response: Thanks for your reminding. We have made changes.

Q7: Subsections of the article: in MPDI format letters are not employed for subsections numbering as for example 1.1, 1.2, etc. Pleae chek the whole manuscript and modify it.

Response: Thanks for your reminding. We have modified the subsections numbering of the whole manuscript to make it more appropriate .

Q8: Figure 6: please, enlarge it.

Response: Thanks for your reminding. We have made changes.

Q9: Table 4: I am not sure if such table is required ori f the information can be summarized in a short paragraph.

Response: Thanks for your reminding. We think that Table 4 can more intuitively compare the differences in the results of several methods.

Round 2

Reviewer 1 Report

Thanks to the authors for considering reviewer's comments and recommendations. In my opinion, now the paper can be accepted.